# Coherent optical control of a superconducting microwave cavity via electro-optical dynamical back-action

Liu Qiu [1,2] ✉, Rishabh Sahu [1,2], William Hease[1], Georg Arnold [1] & Johannes M. Fink[1] ✉

Recent quantum technologies have established precise quantum control of various microscopic systems using electromagnetic waves. Interfaces based on cryogenic cavity electro-optic systems are particularly promising, due to the direct interaction between microwave and optical fields in the quantum regime. Quantum optical control of superconducting microwave circuits has been precluded so far due to the weak electro-optical coupling as well as quasiparticles induced by the pump laser. Here we report the coherent control of a superconducting microwave cavity using laser pulses in a multimode electro-optical device at millikelvin temperature with near-unity cooperativity. Both the stationary and instantaneous responses of the microwave and optical modes comply with the coherent electro-optical interaction, and reveal only minuscule amount of excess back-action with an unanticipated time delay. Our demonstration enables wide ranges of applications beyond quantum transductions, from squeezing and quantum non-demolition measurements of microwave fields, to entanglement generation and hybrid quantum networks.

Microwave superconducting quantum technologies have facilitated the electronic readout and control of superconducting circuits and quantum dot spin qubits[1,2], which holds the promise for quantum-enhanced sensing[3] and scalable quantum computing[4]. Emerging challenges include interfacing the superconducting circuits to complex electrical lines, which introduces excess heat load and complexity beyond traditional cryogenic systems. Photonic fiber links, due to the low propagation loss and passive heating, can be adopted to deliver microwave signals for quantum circuits readout and control at millikelvin temperatures, e.g., using photodiodes[5], mechanical transducers[6,7], or microwave photonics[8,9]. Despite the ubiquitous electro-optic devices in modern telecommunication networks with ultra-high speed translation between electronic and optical fields[10–12], their operations in the quantum regime have been impeded so far due to the weak electro-optical coupling, even at cryogenic temperatures[9].

Cavity electro-optics (CEO) employs resonantly enhanced electro-optic interaction with optimized spatial overlap of microwave and optical modes[13,14]. It holds great promises for general quantum measurement and control of superconducting microwave circuits with optical laser light[14–17], ranging from microwave-optical entanglement generation[18–20], coherent microwave or optical signal synthesis[14], to laser cooling of the microwave mode[21], and bidirectional microwave-optical quantum transduction with near unity efficiency and low added noise[21–24]. A multimode CEO system allows for quantum thermometry[25,26] and quantum non-demolition measurements of the microwave field beyond the standard quantum limit with significantly reduced probing powers[19,27–30]. One particularly promising application of CEO is to build a complex optical quantum network connecting hybrid superconducting microwave quantum circuits[31,32], with alternative approaches using electro- or piezo-optomechanical devices[6,7,33,34], trapped atoms[35,36], rare-earth ions doped crystals[37] and optomagnonic devices[38,39].

Such prospects rely on the optical coherent dynamical control of the superconducting microwave cavity, i.e., via the electro-optical

[1]Institute of Science and Technology Austria, Am Campus 1, 3400 Klosterneuburg, Austria. [2]These authors contributed equally: Liu Qiu, Rishabh Sahu. ✉e-mail: liu.qiu@ist.ac.at; jfink@ist.ac.at

dynamical back-action (DBA)[14]. This has been impeded so far due to the typically weak electro-optical coupling, or the significant excess back-action, i.e., unwanted perturbations that are not due to the electro-optic effect, as a result of the required strong optical pump. Despite the steady progress in the last years, primarily on quantum transductions[21–24], most CEO systems suffer from limited cooperativity $C$[14,40], a measure for coherent coupling versus the microwave and optical dissipation. An endeavor towards coherent electro-optical interaction at unitary cooperativity has started in the last years, including explorations in various electro-optic materials and fabrication processes, e.g., based on aluminum nitride[22,41], bulk and thin-film lithium niobate (LN)[16,21,23,24,42,43], barium titanate[44] and organic polymers[45]. However, excess dissipation[46,47] and back-action still remain in optical and microwave resonators, originating from, e.g., piezoelectric[42,43], photo-refractive effects[48,49], absorption[47], dissipative feedback[50], quasi-particles[45,51], etc.

Pulsed operation in CEO devices reduces the integrated optical power while maintaining the cooperativity, and has recently enabled demonstrations of quantum transduction in the microwave ground state[21,41]. The compatibility of CEO devices to superconducting microwave circuits calls for resolving and controlling pulsed microwave signals in the time domain in a nondestructive manner[4–7]. However, the coherent optical dynamical control of superconducting microwave cavity has remained elusive.

In this work we demonstrate coherent electro-optic dynamical back-action in a multimode cavity electro-optic device. Our results demonstrate coherent stationary and instantaneous electro-optic DBA to the microwave mode, such as the optical spring effect and microwave linewidth narrowing or broadening, with negligible excess back-action. We observe electro-optically-induced absorption or

transparency of the optical-probing field[22,52–54], which opens up the possibility for dispersion engineering of propagating optical and microwave pulses. The observed coherent electro-optical response confirms the feasibility of our multimode CEO system for the direct quantum optical control and sensing of microwave fields in the quantum back-action (QBA) dominant regime[14], and provides important insights into the complex time-dependence of pulsed quantum protocols, e.g., electro-optic entanglement generation[20].

## Results
### Theoretical model and experiment

We realize this experiment in a multimode cavity electro-optical device[16] as depicted in Fig. 1a, where a crystalline lithium niobate whispering gallery mode (WGM) optical resonator is coupled to the azimuthal number $m = 1$ mode of a superconducting aluminum microwave cavity inside a dilution refrigerator at ~10 mK[21,23]. As shown in Fig. 1b, we consider a series of optical transverse-electric (TE) modes of the WGM resonator with the same loss rate $\kappa_o$, i.e., the Stokes, pump and anti-Stokes mode with frequencies $\omega_s$, $\omega_p$, and $\omega_{as}$. When the optical free spectral range (FSR) matches the microwave frequency $\Omega_e$, resonant three-wave mixing between the microwave and adjacent optical modes arises via the cavity enhanced electro-optic interaction, with the interaction Hamiltonian

$$\hat{H}_{eo}/\hbar = g_0 \hat{a}_p^\dagger \hat{a}_s \hat{b} + g_0 \hat{a}_p^\dagger \hat{a}_{as} \hat{b}^\dagger + h.c., \tag{1}$$

where $\hat{a}_s, \hat{a}_p, \hat{a}_{as}$ and $\hat{b}$ are the annihilation operators for the Stokes, pump and anti-Stokes optical and microwave modes, and $g_0$ is the vacuum electro-optical coupling rate. An on-resonance optical pump enhances the electro-optic interaction given by $g = \sqrt{\bar{n}_p} g_0$, where $\bar{n}_p$ is

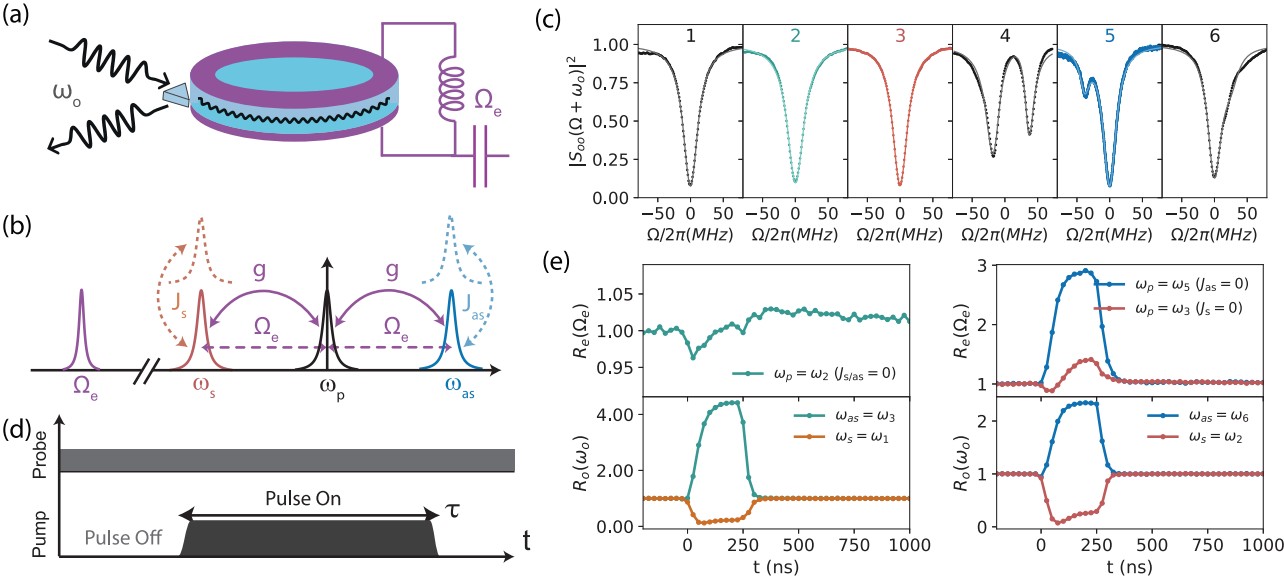

**Fig. 1 | Multimode cavity electro-optical system in the pulsed regime.**
**a** Schematic representation of the cavity electro-optic device. A millimeter-sized lithium niobate optical resonator (light blue) is placed in the capacitor of the LC circuit realized as an aluminum 3D microwave cavity (purple). Optical light is fed to the EO device via an antireflection-coated diamond prism. **b** Mode configurations of the CEO device, with one microwave mode (purple) coupled to three optical TE modes, i.e., the Stokes (red), pump (black) and anti-Stokes mode (blue). A strong optical pump of frequency $\omega_p$ generates pump enhanced Stokes and anti-Stokes scattering at a rate $g$, which can be selectively suppressed by coupling to an optical TM mode (dashed curve). **c** Measured optical reflection (dots) of a series of modes with fitting curves (lines), with mode 2 as the pump mode for the symmetric case ($J_{s/as} = 0$) while mode 3 and 5 for the Stokes ($J_s = 0$) and anti-Stokes ($J_{as} = 0$) case respectively. All resonances are re-centered to the individual TE mode resonance

frequency. Mode splitting in mode 4 indicates strong TE-TM mode coupling.
**d** Coherent dynamical response probing scheme, with a short optical pump pulse of duration $\tau$ and a weak continuous probing field around the microwave or optical (Stokes or anti-Stokes) mode frequency. **e** Temporal on-resonance response $R(\omega)$ [cf. Eq. (3)], i.e., the normalized probing field reflection between pulse on and off (pump peak power ~500 mW). Left panel shows the symmetric case ($\omega_p = \omega_2$), with on-resonance microwave response (green curve) in the upper panel and optical Stokes ($\omega_s = \omega_1$, orange curve) and anti-Stokes ($\omega_{as} = \omega_3$, green curve) responses in the lower panel. Right panel shows the two asymmetric cases, i.e., the on-resonance microwave and optical Stokes responses ($\omega_s = \omega_2$) in the Stokes case ($\omega_p = \omega_3$, red curves), and the on-resonance microwave and optical anti-Stoke responses ($\omega_{as} = \omega_6$) in the anti-Stokes case ($\omega_p = \omega_5$, blue curves).

the mean intra-cavity photon number of the pump mode. This includes the two-mode-squeezing (TMS) interaction between the Stokes and microwave mode [cf. first term in right-hand side of Eq. (1)] and the beam-splitter (BS) interaction between the anti-Stokes mode and microwave mode [cf. second term in right-hand side of Eq. (1)]. One figure of merit of the CEO device is the multiphoton cooperativity $C = 4\bar{n}_p g_0^2/(\kappa_o \kappa_e)$, with $\kappa_o$ and $\kappa_e$ the loss rates of the optical and microwave modes. The TMS or BS interaction can be chosen by selectively suppressing the counterpart via mode engineering, i.e., by coupling the anti-Stokes or Stokes mode to an optical transverse-magnetic (TM) mode of different polarization at rate of $J_{as}$ or $J_s$[16]. The interaction Hamiltonian is given by

$$\hat{H}_J/\hbar = J_s \hat{a}_s^\dagger \hat{a}_{s,tm} + J_{as} \hat{a}_{as}^\dagger \hat{a}_{as,tm} + h.c., \qquad (2)$$

with $\hat{a}_{s,tm}$ and $\hat{a}_{as,tm}$ the annihilation operators for the TM modes of frequency $\omega_s$ and $\omega_{as}$.

Figure 1c shows the optical reflection characterization of one TE mode family of our EO device around 1550 nm with similar total loss rate $\kappa_o/2\pi \approx 26$ MHz. We note that, all modes are re-centered to the individual TE mode resonance. The TE modes are parametrically coupled to a microwave mode with loss rate $\kappa_e/2\pi$ ~10 MHz, whose frequency is adjusted to match the FSR. Mode 4 is strongly coupled to a TM mode of similar frequency with rate $J/2\pi$ ~26 MHz, which manifests as a split mode for anti-Stokes or Stokes scattering suppression when pumping mode 3 or 5, respectively. More details regarding mode characterizations are in the Methods section, including optical losses and mode separations.

In the following we present temporal and spectral coherent dynamical response measurements in the pulsed regime. As shown in Fig. 1d, a strong optical pump pulse of duration $\tau$ is sent to the EO device, together with a weak continuous probing field around the microwave or optical (Stokes or anti-Stokes) resonance, to probe the dynamical back-action during the pulse. We introduce the normalized probing field reflection between pump pulse on and off

$$R_j(\omega) = |S_{jj}(\omega)/S_{jj,off}(\omega)|^2, \qquad (3)$$

with the reflection scattering parameters $S_{jj}(\omega)$, i.e., the output and input field amplitude ratio for mode $j \in (e, o)$.

In Fig. 1e, we show a typical normalized reflection coefficient over time with on-resonance probing in different mode configurations for a pump pulse of duration $\tau = 250$ ns and peak power of ~500 mW. In the symmetric case, i.e., mode 2 as pump mode with $J_{s/as} = 0$, the electro-optical dynamical back-action to the microwave mode is in principle evaded. Due to balanced Stokes and anti-Stokes scattering, the microwave susceptibility remains the same,

$$\chi_e(\Omega) = 1/(\kappa_e/2 - i\Omega). \qquad (4)$$

Interestingly, the optical susceptibilities around the Stokes and anti-Stokes mode frequencies are modified,

$$\chi_{o,s/as}(\Omega) = \frac{1}{\chi_o(\Omega)^{-1} \mp g^2/(\chi_e(\Omega)^{-1} \pm g^2 \chi_o(\Omega))}, \qquad (5)$$

with $\chi_o(\Omega) = 1/(\kappa_o/2 - i\Omega)$ the optical susceptibility. The constructive and destructive interferences between the probing field and the electro-optical interaction result in electro-optically-induced absorption (EOIA) around the Stokes mode and electro-optically-induced transparency (EOIT) around the anti-Stokes mode. Similar dynamics has been reported previously in cavity optomechanics[53,54] and magnomechanics[55], which however only arises in the presence of dynamical back-action[56]. As shown in Fig. 1e (upper left), the microwave on-resonance reflection responds instantaneously to the

arriving pump pulse, and continues to drift even after the pulse is off ($t > 250$ ns). Such excess back-action is negligible, with less than 3% deviation in $R_e(\Omega)$. In Fig. 1e (lower left), the optical on-resonance Stokes (anti-Stokes) reflection decreases (increases) when the optical pulse arrives and restores instantaneously after the pulse is off.

In addition, we consider the Stokes case with mode 3 as pump mode ($J_s = 0$), and the anti-Stokes case with mode 5 as pump mode ($J_{as} = 0$). Coherent electro-optical DBA results in a modified microwave susceptibility,

$$\chi_{e,s/as}(\Omega) = \frac{1}{\chi_e(\Omega)^{-1} \mp g^2 \chi_o(\Omega)}. \qquad (6)$$

DBA on the Stokes (Stokes case) or the anti-Stokes (anti-Stokes case) mode results in the modified susceptibility,

$$\chi_{o,s/as}(\Omega) = \frac{1}{\chi_o(\Omega)^{-1} \mp g^2 \chi_e(\Omega)}, \qquad (7)$$

assuming $4J_{as/s}^2 \gg \kappa_o \kappa_{o,tm}$, with $\kappa_{o,tm}$ the TM mode loss rate. In both cases, Eq. (6) and Eq. (7) are symmetric under interchange of microwave and the optical-probing mode, which enables mutual probing of the optical and microwave field with its counterpart. In the normal dissipation regime, i.e., $\kappa_o \gg \kappa_e$, the microwave mode undergoes effective narrowing (broadening) in the Stokes (anti-Stokes) case, while the Stokes (anti-Stokes) probing field undergoes EOIA (EOIT), due to the constructive (destructive) interference between the probe field and the electro-optical interaction. In the reversed dissipation regime, i.e., $\kappa_o \ll \kappa_e$, the microwave mode experiences EOIA (or EOIT), while the optical Stokes (anti-Stokes) mode linewidth is effectively narrowed (broadened). The temporal on-resonance dynamics in the Stokes and anti-Stokes cases are shown in the right panel of Fig. 1e. Similar to the symmetric case, the Stokes mode undergoes EOIA in the Stokes case, while the anti-Stokes mode undergoes EOIT in the anti-Stokes case.

## Stationary dynamical back-action

As shown in Fig. 1e, the on-resonance normalized reflections remain stationary before and in the middle ($t$ ~ 200 ns) of the pulse. We reconstruct the coherent stationary spectral response by sweeping the probe tone frequency around the probing mode resonance, and perform a pump pulse power sweep in each configuration.

To construct the microwave response, we perform a joint fit of the stationary $R_e(\Omega)$ for different powers, and obtain the individual microwave linewidth and frequency change. The upper panel of Fig. 2a shows the stationary spectral response $R_e(\Omega)$ in three different pump configurations, with the same pump pulse power as in Fig. 1e. $R_e(\Omega)$ remains unchanged due to the balanced Stokes and anti-Stokes scattering in the symmetric case (center), while it changes dramatically around the mode resonance due to strong dynamical back-action in the two asymmetric cases. The lower panel of Fig. 2a shows the measured microwave reflection scattering parameter $|S_{ee}(\Omega)|^2$ with pulse on (off) as solid (dashed) lines, indicating microwave linewidth narrowing and a slight frequency increase in the Stokes case ($\omega_p = \omega_3$) and linewidth broadening in the anti-Stokes case ($\omega_p = \omega_5$) with an increased on-resonance reflection.

In Fig. 2b, we show the extracted microwave frequency ($\delta\Omega_e$) and linewidth ($\delta\kappa_e$) change in the power sweep, for each pump configuration. The corresponding microwave response fitting curves are shown in Figs. S7–S10 in SI. In the symmetric case ($\omega_p = \omega_2$), no evident frequency or linewidth change is observed due to the evaded back-action. In the anti-Stokes case ($\omega_p = \omega_5$) the microwave linewidth increases linearly with $C$, while it decreases in the Stokes case ($\omega_p = \omega_3$). The theoretical curves for both asymmetric cases match very well with experimental results, using a full dynamical back-action model

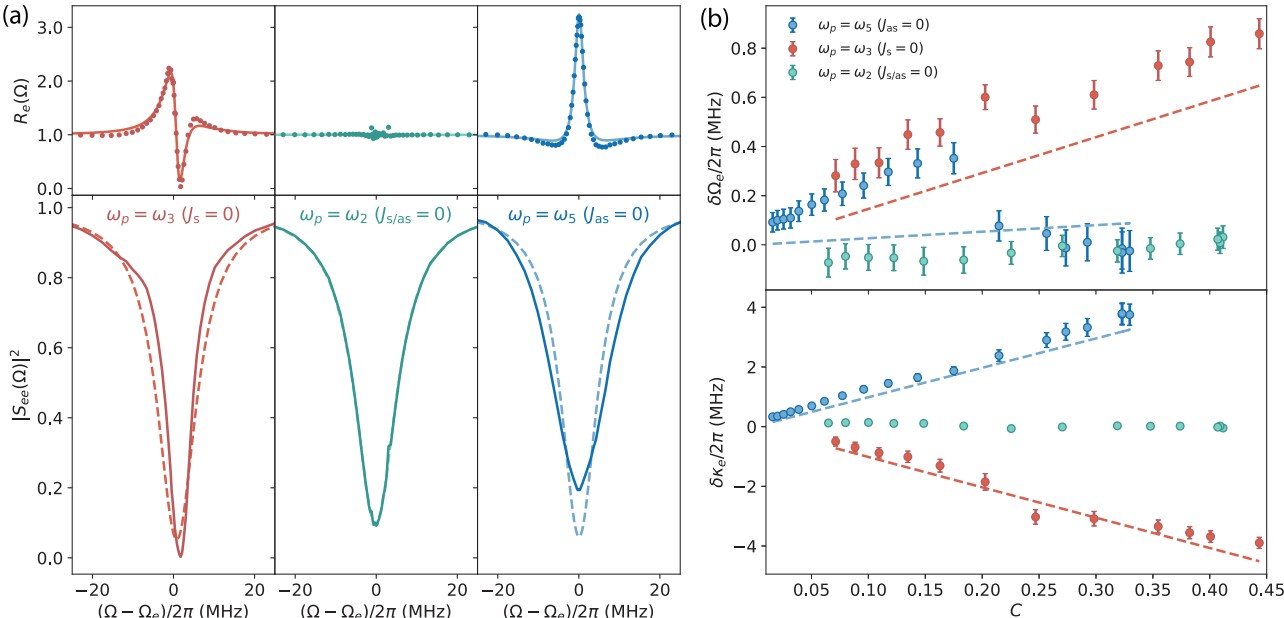

**Fig. 2 | Stationary dynamical back-action to the microwave mode.** A power sweep is conducted in each pump configuration. A joint fit of the stationary $R_e(\Omega)$ is performed with the original microwave linewidth as a shared parameter, and the linewidth and frequency change for each power as remaining fitting parameters. **a** Microwave response measurements with the same pump power as in Fig. 1e. The upper panel shows the stationary $R_e(\Omega)$ as dotted lines, with fitting curves as solid lines. The lower panel shows the reconstructed microwave reflection $|S_{ee}(\Omega)|^2$ with the pump on (solid curve) and off (dashed curve) using obtained parameters from the joint fit. **b** Fitted microwave frequency shift and linewidth change versus cooperativity $C$. Dashed lines are theoretical curves incorporating the full dynamical back-action model, using fitting parameters from the corresponding coherent optical response [cf. Fig. 3], including imperfect frequency detunings. Error bars represent the 95% confidence interval of the fit.

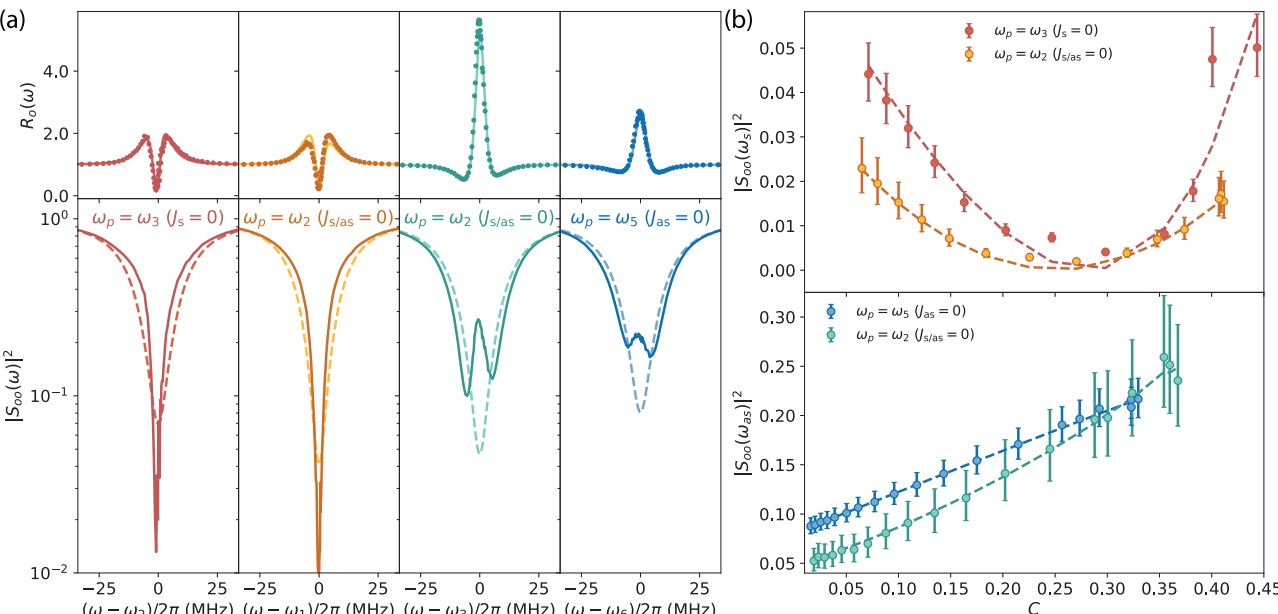

**Fig. 3 | Stationary electro-optically-induced absorption and transparency, with Stokes mode probing in the Stokes and symmetric cases, while anti-Stokes mode probing in the symmetric and anti-Stokes cases.** In each probing configuration, a pump power sweep is conducted, and a joint fit of stationary $R_o(\omega)$ to the full dynamical back-action model is performed. **a** Measurements with same pump power as in Fig. 1e, with the two left panels for Stokes mode probing and the two right panels for anti-Stokes mode probing. The upper panel shows $R_o(\omega)$ (dotted lines) with fitting curves (solid lines). The lower panel shows reconstructed optical reflection $|S_{oo}(\omega)|^2$ with pulse on (solid curve) and off (dashed curve) in logarithmic scale, which demonstrates EOIA in the Stokes case and EOIT in the anti-Stokes case. **b** The upper panel shows $|S_{oo}(\omega_s)|^2$ for the two Stokes mode probing cases, while the lower panel shows $|S_{oo}(\omega_{as})|^2$ for the two anti-Stokes mode probing cases. The corresponding theoretical curves are shown as dashed lines. Error bars indicate two standard deviations.

incorporating optical response fitting parameters including imperfect frequency detunings [cf. Fig. 3b]. In the anti-Stokes case, we observe a minuscule deviation in the microwave frequency shift of ~$10^{-4}\Omega_e$. This can be explained by the small detuning uncertainties (sub-MHz) as

discussed in SI, probably due to photo-refractive[48,49] or quasi-particles effects[6,51].

As shown in the upper panel of Fig. 3a, we perform a joint fit of the stationary $R_o(\omega)$ in each probing configuration, i.e., Stokes mode

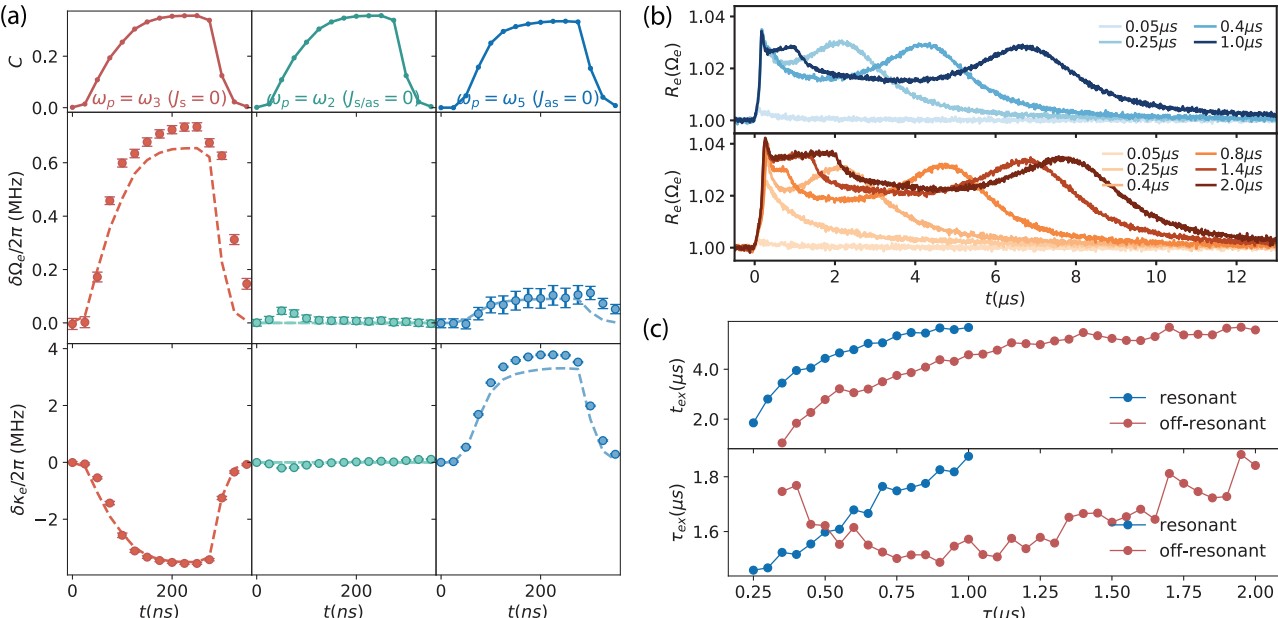

**Fig. 4 | Transient dynamical back-action to the microwave mode.**
**a** Instantaneous coherent microwave responses during the pump pulse obtained for the same pump power as in Fig. 1e. In each pump configuration, we perform a joint fit of $R_o(\omega)$ over the time of the pulse with the full dynamical back-action model, and extract the time dependent $C(t)$ (upper panel). Similarly, we perform a joint fit of $R_e(\Omega)$ to obtain the frequency and linewidth changes (dotted lines) over time, with corresponding theoretical curves (dashed lines) using the above fitted parameters from the optical coherent response. Error bars represent the 95% confidence interval of the fit. **b** Delayed excess back-action to the microwave mode in the resonant condition ($\Omega_e = $ FSR, upper panel) and the off-resonant condition ($\Omega_e \neq $ FSR, lower panel), in the symmetric mode configuration. Different curves correspond to different pulse lengths for a constant pump power similar to Fig. 1e. Delayed excess back-action still exists after the pulse, and results in a bounce in $R_e(\Omega_e)$ after $t_{ex}$ of several μs, which then decreases exponentially to 1 with the time constant $\tau_{ex}$. **c** Extracted bounce time $t_{ex}$ after the pulse is off and the mean decay time $\tau_{ex}$ as a function of pulse length. The blue and red curves correspond to the resonant and off-resonant cases.

probing in the Stokes and symmetric cases while anti-Stokes mode probing in the symmetric and anti-Stokes cases. This allows us to extract $C$, $\kappa_o$, and external coupling rate $\kappa_{o,ex}$ in each probing configuration. In the lower panel of Fig. 3a, we show the reconstructed optical reflection efficiency $|S_{oo}(\omega)|^2$ with pulse on and off as solid and dashed lines. The Stokes mode probing (left two panels) reveals similar EOIA for the Stokes and symmetric cases when the pump pulse is on, while the anti-Stokes mode probing (right two panels) indicates similar EOIT for the symmetric and anti-Stokes cases. In Fig. 3b, we show the on-resonance reflection efficiency versus $C$ in different probing configurations with theoretical curves shown as dotted lines. In the upper panel, $|S_{oo}(\omega_s)|^2$ at the Stokes mode resonance first approaches zero and then increases with $C$ due to EOIA. In the lower panel, $|S_{oo}(\omega_{as})|^2$ at the anti-Stokes resonance increases slowly as $C$ increases due to EOIT. We note that, the different on-resonance $|S_{oo}|^2$ at low $C$ is due to the slightly different external coupling efficiency of the optical modes. To capture the stationary electro-optical dynamics, the effective $C$ is limited to ~0.5 due to the Kerr nonlinearity[21], which depends on the power and duration of the applied pulse and results in optical parametric oscillation in the optical resonator[57]. With further improvement of $\kappa_e$ and $g_0$, the device can enable parametric amplification of the microwave and optical Stokes signal for $C \gg 1$.

**Transient dynamical back-action**
Emerging quantum applications of CEO devices, such as ultra-low noise microwave-optical quantum transduction and entanglement generation, require strong optical pump pulses to reach near unity $C$[21,24]. A detailed understanding of the transient response of CEO devices is therefore crucial for complex measurement protocols in the quantum limit.

In Fig. 4a, we show the transient response of the microwave mode in different pump configurations with the same power as in Fig. 1e.

Within each pump configuration, we perform a joint fit of $R_o(\omega)$ over the pulse incorporating the full DBA model, with $C(t)$ and imperfect detunings as free parameters, as explained in SI. When the optical pump pulse arrives, the fitted $C(t)$ increases smoothly in the beginning, reaches stationary value in the middle, and slowly decreases to zero after the pulse. In the middle and lower panel, we show the obtained microwave frequency and linewidth change over the pulse as dotted lines, with theoretical curves as dashed lines. The small blue shift of the microwave mode in the two asymmetric mode configurations is due to imperfect detunings (sub-MHz). The linewidth change follows closely the predicted coherent electro-optical dynamical back-action, i.e., narrowing in the Stokes case while broadening in the anti-Stokes case. In the symmetric case, a very slight excess frequency drift (~$10^{-5}\Omega_e$) and linewidth change (~$10^{-2}\kappa_e$) indicate a finite amount of instantaneous excess back-action to the microwave mode in the beginning and at the end of the pulse, due to the loading and unloading of the optical pump field. We note that, similar instantaneous excess back-action also appears in the optical coherent response, which results in an estimated detuning jiggle (sub-MHz) during the pulse, as shown in Fig. S5 in SI.

**Delayed excess back-action**
After the pump pulse, a small amount of excess back-action remains in the microwave mode for a few μs, while it ceases immediately in the optical mode. The details of the extracted microwave frequency and linewidth drift at different $C$ over time in the symmetric case [cf. Fig. 4a] are discussed in Fig. S6 in SI. In Fig. 4b, we show instead a comparison between two different resonant conditions, i.e., the resonant case ($\Omega_e = $ FSR) and the off-resonant case ($\Omega_e \neq $ FSR, by detuning the microwave resonance frequency), in the symmetric mode configuration ($\omega_p = \omega_2$). In the upper panel, we show $R_e(\Omega_e)$ over time using a similar pulse power as in Fig. 1e, for different pulse lengths $\tau$. The off-resonant case shown in the lower panel ($\Omega_e - $ FSR $= 2\pi \times 100$ MHz)

results in a similar microwave response, which rules out the electro-optical interaction as the main origin of the induced perturbation. For short pump pulses (e.g., below 200 ns), $R_e(\Omega_e)$ decreases right after the pulse and restores slowly to unity in both cases. For longer pulses (e.g., above 300 ns), $R_e(\Omega_e)$ reveals an unanticipated delayed back-action, which decreases in the beginning and exhibits a bounce around $t_{ex}$ (several µs) after the pulse. As the pulse length increases, $t_{ex}$ increases accordingly, which indicates an integrated optical pulse energy dependent excess mechanism that changes the microwave response—predominantly the mode frequency as explained in SI A and E—as corroborated by a pump power sweep with similar results [cf. Fig. S6 in SI]. After the bounce, $R_e(\Omega_e)$ continues to decrease exponentially to unity with a time constant $\tau_{ex}$. In Fig. 4c, we show the extracted $t_{ex}$ and $\tau_{ex}$ from the fitted time dependence for different pump pulse lengths as shown in Fig. S11. In cases, $t_{ex}$ increases versus pulse length $\tau$ and saturates at ~6 µs for long pulse lengths above ~1 µs, while the resonant excess back-action arrives later than the off-resonant one most likely related to electro-optical interaction. While the fitted decay time $\tau_{ex}$ exhibits slightly different relations to the pulse length in both cases, the fitted value is quite similar ( ~1.6 µs), indicating a general underlying mechanism, which requires further investigation, e.g., light-induced quasi-particles[6,51] or photo-refractive effects[48,49].

It is important to point out that the observed frequency shifts and linewidth changes are only on the order of 100 kHz, i.e., $10^{-5}$ of the microwave resonant frequency. Moreover, our CEO device revives completely only tens of µs after the pulse. Nevertheless, in all the presented experiments we adopt a repetition time of 10 ms for all pump configurations (500 ms in the Stokes case) to avoid optical heating. Both stationary and transient coherent dynamics are independent from the different repetition rates. We note that, the low repetition rate is important to remain in the quantum back-action dominated regime for microwave-optics entanglement generation[20].

## Discussion

We have demonstrated coherent optical control of a microwave cavity in the pulsed regime in a multimode EO device at millikelvin temperature with near-unitary cooperativity. Both the stationary and instantaneous response of the microwave and optical-probing field agree very well with coherent DBA theory, except for a very small and for many applications negligible excess back-action that is in fact surprisingly small given the large optical photon energy compared to the small superconducting gap of aluminum.

The presented coherent optical control of a superconducting microwave cavity mode confirms the compatibility between optical light and superconducting microwave circuits in our device, and enters a new strong interaction regime of quantum electro-optics. It also paves the way for a wide range of quantum applications beyond microwave-optical transduction[21,39], ranging from entanglement generation[20], optically driven masing and squeezing, to quantum thermometry[26] and precision measurement of microwave radiation beyond the standard quantum limit[28]. Our EO device offers great compatibility to cryogenic microwave circuits, and represents a promising platform for the generation of non-classical microwave-optical correlations to realize a distributed quantum network between superconducting quantum processors[4,31].

## Methods
### Device characterization
The multimode CEO device used in the reported experiments is the one also used in refs. 21,23. It consists of a millimeter-sized LiNbO₃ whispering gallery mode resonator (WGMR) and a superconducting aluminum cavity at mK stage in a Bluefors dilution refrigerator. The optical modes with optimal TE mode coupling are characterized individually using laser piezo scanning, whose normalized reflection are shown in Fig. 1c with Lorentzian fit, e.g., for mode 1, 2, and 3. The

optical resonances are ~1550 nm, with similar fitted total linewidth of $\kappa_o/2\pi$ ~ 26 MHz, of which the external coupling rate is $\kappa_{o,ex}/2\pi$ ~ 10 MHz. We note that, light couples to the WGMR via a diamond prism, with imperfect spatial field mode overlap $\Lambda = 0.83$. For simplicity, throughout our work, we include the effective $\Lambda$ factor in $\kappa_{o,ex}$. For mode 4, 5 and 6, coupling between the TE and TM mode results in mode splitting or distortion. The mode with largest splitting, i.e., mode 4, is adopted as the split mode in the asymmetric case. Due to the slight splitting in mode 5, effective $C$ is slightly reduced, as evident in Fig. 4a with same pump pulse power. Because of the large frequency difference and relative weak coupling between the TE and TM mode in mode 6, we approximate it as a single TE mode in the main text. Depending on the specific pump configuration, the microwave cavity frequency is adjusted to match the optical pump and probe mode separation. The microwave mode has similar total loss rate $\kappa_e/2\pi$ ~ 10 MHz with external coupling rate $\kappa_{e,ex}/2\pi$ ~ 4 MHz.

### Data analysis
The spectral normalized reflection for the probing field is given by,

$$R_j(\omega) = |S_{jj}(\omega)/S_{jj,\text{off}}(\omega)|^2, \tag{8}$$

where $S_{jj}(\omega)$ and $S_{jj,\text{off}}(\omega)$ are the reflection coefficient ($S_{11}$ parameter) of the probing field $j$ with pulse on and off. In the absence of the pump pulse, the output photon number of the probing field takes the form

$$\bar{n}_{\text{out,off}}(\omega) = \bar{n}_{\text{in}}(\omega)|S_{jj,\text{off}}(\omega)|^2 \eta_d(\omega), \tag{9}$$

where $\bar{n}_{\text{in}}(\omega)$ and $\eta_d(\omega)$ are the frequency dependent input photon number and the detection efficiency. After the pump pulse arrives, the output photon number of the probing field is modified to,

$$\bar{n}_{\text{out}}(\omega) = \bar{n}_{\text{in}}(\omega)|S_{jj}(\omega)|^2 \eta_d(\omega). \tag{10}$$

For long repetition time as in our experiments, the coherent response of the probing field restores to the state before the pulse starts, where we approximate $S_{jj,\text{off}}(\omega)$ to $S_{jj}(\omega)|_{t=0}$.

In the experiments, the weak coherent RF signal from the down-converted microwave or optical field $I_j(t)$ is located at 40 MHz, more than 10 dB above the noise floor, due to the low noise amplification using HEMT amplifier or optical balanced heterodyne detection. We perform digital down-conversion of the time-domain data at 40 MHz for each probing frequency, where the averaged voltages over the pulses are adopted to obtain the mean power. We can track the normalized reflection coefficient over time by scanning the probe field frequency,

$$R_j(\Omega + \Omega_{\text{LO}j}) = \frac{\bar{P}_{\text{out},j}(\Omega)}{\bar{P}_{\text{out},j}(\Omega)|_{t=0}}, \tag{11}$$

with $\Omega_{\text{LO}j}$ the LO frequency and $\bar{P}_{\text{out},j}$ the averaged power of the RF field from digital down-conversion. Typical obtained on-resonance $R_j(\omega)$ in time domain are shown in Fig. 1e. In this way, we avoid complicated calibrations of the frequency dependent input and detection lines, and reconstruct the reflection coefficient $S_{jj}(\omega)$ by fitting $R_j(\omega)$ to theoretical models as detailed in SI.

## Data availability
The data that support the findings of this study have been deposited in a Zenodo repository with https://doi.org/10.5281/zenodo.7936405.

## Code availability
The code used in this study for data analysis and modeling has been deposited in a Zenodo repository with https://doi.org/10.5281/zenodo.7936405.

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

## Acknowledgements

This work was supported by the European Research Council under grant agreement no. 758053 (ERC StG QUNNECT), the European Union's Horizon 2020 research and innovation program under grant agreement no. 899354 (FETopen SuperQuLAN), and the Austrian Science Fund (FWF) through BeyondC (F7105). L.Q. acknowledges generous support from the ISTFELLOW programme. W.H. is the recipient of an ISTplus postdoctoral fellowship with funding from the European Union's Horizon 2020 research and innovation program under the Marie Skłodowska-Curie grant agreement no. 754411. G.A. is the recipient of a DOC fellowship of the Austrian Academy of Sciences at IST Austria.

## Author contributions

L.Q. conceived the idea for the experiment. L.Q. and R.S. performed the experiments together with W.H. and G.A. L.Q. developed the theory and performed the data analysis. The manuscript was written by L.Q. with assistance from all authors. J.M.F. supervised the project.

## Competing interests

The authors declare no competing interests.
