## [Peer Review File · Nature Communications]

REVIEWER COMMENTS

Reviewer #1 (Remarks to the Author):

This manuscript experimentally studies the effect of an optical pump pulse on a weak microwave or optical probe. Overall, I think this manuscript is a comprehensive study of the dynamics of the microwave and optical probe in the presence of a high-power optical pump. The authors demonstrate how the microwave mode and optical mode evolve in the presence of an optical pump pulse. As far as I am aware of his work is the first to comprehensively study the effect of pump pulse in an electro-optic cavity and demonstrate various known effects (EIT, EIA, microwave resonance narrowing/broadening, etc). Their experiments and theoretical fittings agree well. The drawback is that results shown here are not fundamentally new. This manuscript lies at the boundary of acceptance in terms of novelty and device advances.

Reviewer #2 (Remarks to the Author):

The manuscript by Qiu and Sahu et al. is a culmination of a series of work to develop a (bulk) electro-optic microwave–optical transducer, motivated by the vision of building long-haul optical interconnects for superconducting qubits that operate in the microwave regime. In [1], the first version of the device was first proposed and characterized at room temperature. The work was then followed by integration with a superconducting microwave resonator and demonstration of bidirectional conversion in the quantum ground state [added (microwave) output noise $\ll 1$] at dilution refrigerator temperature [2]. In 2021, a landmark experiment [3] conducted by the authors attained "quantum-enabled" operation (added noise referred to the input < 1)—first among electro-optic transducers to my knowledge—by leveraging pulsed pumping to achieve near-unity peak electro-optic cooperativity. Therein evidence for direct electro-optic laser cooling and vacuum amplification was also presented. In this manuscript, the authors have gone one step further and present a beautiful demonstration of electro-optic dynamical backaction phenomena, by virtue of the Pockels Hamiltonian taking on the same form as that of cavity optomechanics. The work is certainly a significant experimental undertaking, with considerable impact in both quantum networking and microwave photonics. While the work is suitable for Nature Communication in both quality and impact in my opinion, there are several points that I'd like to see addressed:

[1] Rueda et al., *Optica* 3, 597-604 (2016)

[2] Hease et al., *PRX Quantum* 1, 020315 (2020)

[3] Sahu et al., *Nat Commun* 13, 1276 (2022)

- Fig. 2b (and less strikingly, Fig. 4a): The theory-data discrepancy is attributed to the sub-MHz "detuning uncertainties", but the apparent discontinuity in the data that occurs around $C = 0.2$ is concerning. A couple sentences following Line 264 would be very helpful. Was the order of data acquisition randomized with respect to C ? Aside from providing some physical intuition on the discontinuity (photorefractive effect?), could the authors include e.g. a simulation in the SI to give the reader a sense of how this detuning could engender such a deviation in this parameter regime?

- Fig. 1e (top left panel) vs Fig. 4b (upper panel): I am puzzled by the apparent contradiction between the plotted data in these two panels. If I understood correctly, the same pump power and pump configuration (symmetric) are applied in the two cases. In Fig. 1e, it can be seen that the normalized microwave reflection dips below 1, whereas in Fig. 4b, the normalized reflection appears to be always above 1 for a similar pulse length (~ 200 ns).

- The data analysis to extract dynamical backaction parameters rely heavily on the "joint fit" procedure. While the authors have supplied additional data and fit results in Fig. S6–S9, I personally find this a little opaque. It would be very helpful to provide an illustrative example in the SI, holding the reader's hands to go through the analysis of one dataset step by step.
- Line 366 / Fig. 4c: Based on the data and uncertainties provided here, I'd venture to argue that the fitted decay time exhibits different trends in the two detuning cases. In the resonant case, the decay time increases monotonically as a function of pulse length (which one might expect), whereas the off-resonance case exhibits a minimum around a pulse length of 1 μ s. I am aware that the y-axis range is quite narrow, but perhaps additional explanation is warranted here.
- Line 291: It is mentioned that here the effective cooperativity achievable is limited to ~ 0.5 due to Kerr nonlinearity, whereas in the authors' previous paper [3] cooperativity ~ 1 was reached. Is it because of the longer pulses used here for the purpose of obtaining the stationary response? One potentially useful piece of information to report would be the cooperativity threshold as a function of pulse length. As the authors mention in Line 397, an optically driven maser would indeed be a very useful device; it'd be even more useful if such a maser could operate in a regime of high duty cycle. The data could help the readers access the potentials and limitations of the device in the context of microwave photonics.
- Line 375: It may be useful to have a one-sentence explanation on why the Stokes configuration requires a different triggering setting (lower duty cycle).

Finally, some comments and suggestions on semantics / notation / general writing:

- Line 284: The sentence is worded in a somewhat misleading fashion. My understanding is that no data with $C \gg 1$ are provided in the manuscript. The authors may want to refer the reader to their simulation in the SI (Fig. S1) to avoid potential confusion.
- The notation for various frequencies appear to be used interchangeably from time to time: e.g., Fig. 1c (Ω = optical detuning from each FSR mode), Fig. 2a (Ω = microwave frequency). Although optical detuning is indeed in the GHz regime, my understanding is that the authors intend to reserve Ω (ω) for microwave (optical) frequencies. It would be good to preserve this consistency.
- I noticed that some figures are plotted such that data points are cropped out (e.g., Fig. 3b, Fig. 4a). The authors may want to re-adjust the axis limits to include these points fully.
- "Trigger time" is mentioned several times throughout the main text (e.g. Line 375) and the SI. There may be a technical reason for using this term, but I personally find "cycle time" or "repetition time (rate, if quoted in frequency)" to be a little more illuminating.
- Line 400: I am aware that the authors are not claiming to have reached the "quantum-limited regime" in this manuscript and the sentence is meant to be a projection, but I would clearly indicate this as such. In fact, in their previous paper [3], efforts were made to distinguish between "quantum-limited" and "quantum-enabled".
- Line 394: I imagine many people from the circuit QED community may read this paper. Therefore, it may be helpful to clarify that the "strong interaction regime of microwave photonics" refers to optical generation of microwaves, rather than something involving qubits [e.g. Ma, R., Saxberg, B., Owens, C. et al. Nature 566, 51–57 (2019).]
- Line 292: I think "Kerr the nonlinearity" should be "Kerr nonlinearity".
- I may not be the most qualified to comment on English, but I would use "resonant" ("off-resonance") to replace all the "on-resonant" ("off-resonant").

Reviewer #3 (Remarks to the Author):

The authors present results characterizing the electro-optic interaction between a lithium niobate optical resonator and a microwave cavity. They demonstrate phenomena such as

EOIA and EOIT, along with dynamical backaction effects, whose analogies were previously mostly observed in optomechanics, but are now accessible accessible in their electro-optic device, thanks to a cooperativity that approaches unity. Furthermore, they present some less well-understood observations, for example a delayed effect of the optical pump on the response of the microwave cavity.

The experimental techniques analysis seem sound, and the results are carefully presented. Given the potential applications of this system, such as quantum transduction, a manuscript presenting the details of its dynamics is certainly useful. However, I am a bit puzzled by the context of this work with respect to previous publications using this device, for example ref. 20 and 22. It seems like the device used here is similar to the one used in those works. The introduction, however, makes it sound like this is a new regime of CEO systems, but the cooperativities shown here were already demonstrated in ref. 20 for example. If there is indeed some new technical development, the authors should make it clearer. If not, the results in this manuscript could have perhaps been included in the supplement to ref. 20 or 22. Though I'm not against publishing a separate paper (but perhaps in a more specialized technical journal) instead of putting all this in a supplement, the relationship to the authors' earlier works should be made clear so as to not confuse the reader.

Some other specific comments are:

1. Under the category of "trapped atoms" on line 57, the authors could consider citing the recent paper by Kumar et al (arXiv:2207.10121).
2. The authors mention "excess back-action" multiple times in the abstract and intro without explaining what this means. Without going into what exact physical process leads to this back-action, they could at least briefly state what sorts of phenomena they're referring to (heating, additional loss, etc.)
3. I found the labeling of "Stokes" and "anti-Stokes" a bit confusing sometimes. In many cases, it would have been more clear if the authors specified where the probe field was located. For example, in the caption to Fig. 1e, when the authors say the "optical Stokes (orange curve) and anti-Stokes (green curve) responses", I assume they mean the probe response around mode 1 and 3?
4. Could the authors comment on the shape of the red curve in the upper right panel of Fig 1e? I guess it's due to some combination of the time-varying narrowing and frequency shift of the mode?
5. In the caption to Fig. 2a, it says "the lower panel shows the reconstructed microwave reflection..." What does "reconstructed" mean?
6. In the discussion of Fig. 2b, the authors say that the deviation of the frequency change is due to small detuning uncertainties. Do they mean the deviation of the data from the theory curve? I guess the theory curve includes the effect of detuning (according to SI A), so does "uncertainty" mean an unknown detuning in the measurement that also seems to vary with cooperativity?
7. In line 375, by "trigger time", do the authors mean repetition period of the experiment?

Reply to Referee's Comments on Manuscript NCOMMS-22-51829-T

The original referee reports are shown in black. Our replies are shown in blue.

Reviewer #1 (Remarks to the Author):

This manuscript experimentally studies the effect of an optical pump pulse on a weak microwave or optical probe. Overall, I think this manuscript is a comprehensive study of the dynamics of the microwave and optical probe in the presence of a high-power optical pump. The authors demonstrate how the microwave mode and optical mode evolve in the presence of an optical pump pulse. As far as I am aware of his work is the first to comprehensively study the effect of pump pulse in an electro-optic cavity and demonstrate various known effects (EIT, EIA, microwave resonance narrowing/broadening, etc). Their experiments and theoretical fittings agree well. The drawback is that results shown here are not fundamentally new. This manuscript lies at the boundary of acceptance in terms of novelty and device advances.

We are glad that the reviewer also identifies our results as the first comprehensive study to demonstrate coherent optical control of a superconducting resonator. In addition, we appreciate the reviewer's acknowledgement of the excellent agreement between our experimental results and the theoretical model.

The main concern is regarding the novelty of our results. We note that, since the original theoretical proposal by Tsang [Tsang, PRA 2010], all the experiments in the field are only focused on the problem of **microwave-optical conversion**. However, our experiments demonstrated **for the first time** the long-sought-after physics, *i.e. coherent electro-optical dynamical back-action, the essence of quantum electro-optics*, which has been anticipated for more than one decade. This constitutes a defining step in the field of cavity electro-optics, similar to the outset of quantum optomechanics, and entails the readiness of the device for exploration of a rich set of exciting physical phenomena and applications in quantum networks and microwave photonics **beyond quantum transduction**.

In addition, our results *cease the long-lasting concern in the community regarding **the compatibility between optical light and superconducting cavities***. Over the last years, various research groups have *repeatedly shown* that excess microwave losses (Cooper pair breaking by high-energy optical photons) and substantial optical losses (due to high conductivity superconductors) overshadow any coherent interaction. Our first demonstration of coherent electro-optical control *provides further insights to alternative platforms (also improvement of our own device)* with similar challenges in *fabrication, material science, and device designs*.

As suggested by the reviewer, we have revised the abstract, introduction and conclusion to elaborate further on the novelty of our results in the revised version, e.g.

- 1) in the abstract, "...using laser light beyond microwave-optical quantum transduction, with possible applications ... and squeezing, to entanglement generation..."
 - 2) in the conclusion, "The presented coherent optical control of a superconducting microwave cavity mode confirms the compatibility between optical light and superconducting microwave circuits, and enters a new strong interaction regime of quantum electro-optics. It also... quantum applications beyond microwave-optical transduction, ranging from entanglement generation, ...".
-

Reviewer #2 (Remarks to the Author):

The manuscript by Qiu and Sahu et al. is a culmination of a series of work to develop a (bulk) electro-optic microwave–optical transducer, motivated by the vision of building long-haul optical interconnects for superconducting qubits that operate in the microwave regime. In [1], the first version of the device was first proposed and characterized at room temperature. The work was then followed by integration with a superconducting microwave resonator and demonstration of bidirectional conversion in the quantum ground state [added (microwave) output noise $\ll 1$] at dilution refrigerator temperature [2]. In 2021, a landmark experiment [3] conducted by the authors attained "quantum-enabled" operation (added noise referred to the input < 1)—first among electro-optic transducers to my knowledge—by leveraging pulsed pumping to achieve near-unity peak electro-optic cooperativity. Therein evidence for direct electro-optic laser cooling and vacuum amplification was also presented. In this manuscript, the authors have gone one step further and present a beautiful demonstration of electro-optic dynamical backaction phenomena, by virtue of the Pockels Hamiltonian taking on the same form as that of cavity optomechanics. The work is certainly a significant experimental undertaking, with considerable impact in both quantum networking and microwave photonics. While the work is suitable for *Nature Communication* in both quality and impact in my opinion, there are several points that I'd like to see addressed:

[1] Rueda et al., *Optica* 3, 597-604 (2016)

[2] Hease et al., *PRX Quantum* 1, 020315 (2020)

[3] Sahu et al., *Nat Commun* 13, 1276 (2022)

We highly appreciate the acknowledgement regarding the quality and impact of our manuscript as "*significant experimental undertaking, with considerable impact in both quantum networking and microwave photonics*" with the demonstration electro-optical dynamical back-action phenomena. Indeed, our experiment *ushers in a new paradigm of cavity electro-optics, i.e. a general platform for quantum optical manipulation of microwaves.*

- Fig. 2b (and less strikingly, Fig. 4a): The theory-data discrepancy is attributed to the sub-MHz "detuning uncertainties", but the apparent discontinuity in the data that occurs around $C = 0.2$ is

concerning. A couple sentences following Line 264 would be very helpful. Was the order of data acquisition randomized with respect to C ? Aside from providing some physical intuition on the discontinuity (photorefractive effect?), could the authors include e.g. a simulation in the SI to give the reader a sense of how this detuning could engender such a deviation in this parameter regime?

We thank the reviewer for the comment. Indeed, there is a theory-data discrepancy regarding the microwave frequency change, especially around $C=0.2$, in Fig.2b and Fig.4a. The data acquisition here is done with pumping power reduced over time (thus not randomized with respect to C). The total measurement time of a single blue curve in Fig.2(b) is around 9 hours, considering the sweeping of the pump powers and microwave/optical probing frequencies. The discrepancy might result from a slow drift of the device parameters, due to e.g. photorefractive effect on the optical side or quasiparticles on the microwave side. For simplicity in the fitting, we assume the same microwave resonant frequency for the entire power sweep measurements. We note that, such microwave frequency change discrepancy (sub-MHz) is negligible compared to microwave frequency ~ 9 GHz (line 262 in the manuscript).

As suggested by the reviewer,

- 1) we have added a discussion regarding the discrepancy after line 263.
- 2) we have provided an additional figure in the SI (Section A) to better elaborate on how the microwave frequency changes (optical spring effect) to the imperfect frequency detunings.

- Fig. 1e (top left panel) vs Fig. 4b (upper panel): I am puzzled by the apparent contradiction between the plotted data in these two panels. If I understood correctly, the same pump power and pump configuration (symmetric) are applied in the two cases. In Fig. 1e, it can be seen that the normalized microwave reflection dips below 1, whereas in Fig. 4b, the normalized reflection appears to be always above 1 for a similar pulse length (~ 200 ns).

We thank the reviewer for the comment regarding the contradiction between the two plots. The change in microwave reflection is a consequence of a simultaneous microwave cavity frequency shift and a linewidth change. In Fig. 1e, the MW reflection decreases upon the arrival of the strong optical pump pulse. This is mostly due to the microwave linewidth decrease, as evident in Fig. 4(a) (middle bottom) in the main text, despite of the symmetric pump case. In Fig. 4(b,c), we performed similar experiments at higher pump power using *a spectrum analyzer in the "zero-span mode", i.e. at a single frequency bin* for better signal to noise. The measurement frequency might be slightly *different* from the microwave resonance. This is in contrast to measurements in Fig. 4(a), where *a frequency sweep is performed while using a digitizer for data collection*.

For better clarity, we have updated Fig. S5(b) in the SI, where we have included the region with the optical pump pulse on, which clearly shows the microwave linewidth decrease at high C when the pulse arrives. In addition, we have added additional discussions regarding the microwave reflection decrease (Fig. 1e) in Section E of the SI.

- The data analysis to extract dynamical backaction parameters rely heavily on the "joint fit" procedure. While the authors have supplied additional data and fit results in Fig. S6–S9, I personally find this a little opaque. It would be very helpful to provide an illustrative example in the SI, holding the reader's hands to go through the analysis of one dataset step by step.

We would like to point out that the data analysis procedure is discussed in quite some detail in Section C of the SI. This section includes, 1) the basic principle for susceptibility reconstruction, 2) basic fitting functions for microwave (S23) and optics (S10, S13 and S17), 3) shared and independent parameters for the joint fit. Besides, as we mentioned in the data availability statement, we will provide both code and raw data on the Zenodo repository upon acceptance. For better clarity, we have also reorganized the Section C with several subsections.

- Line 366 / Fig. 4c: Based on the data and uncertainties provided here, I'd venture to argue that the fitted decay time exhibits different trends in the two detuning cases. In the resonant case, the decay time increases monotonically as a function of pulse length (which one might expect), whereas the off-resonance case exhibits a minimum around a pulse length of 1 μ s. I am aware that the y-axis range is quite narrow, but perhaps additional explanation is warranted here.

We thank the reviewer for this comment. Indeed, the decay time in both cases show different relations to the pulse length. In addition, as pointed out and also mentioned in the manuscript, the fitted decay time is actually quite close, e.g. $\sim 1.6 \mu$ s. This may imply a general underlying mechanism which requires further investigation, e.g. quasiparticles or photorefractive effects.

To avoid confusion, we have rephrased the sentence in line 371: "While the fitted decay time τ_{ex} exhibits a slightly different relation to the pulse length in both cases, the fitted value is quite similar ($\sim 1.6 \mu$ s), indicating a general ..., which requires further investigation, e.g. ..."

- Line 291: It is mentioned that here the effective cooperativity achievable is limited to ~ 0.5 due to Kerr nonlinearity, whereas in the authors' previous paper [3] cooperativity ~ 1 was reached. Is it because of the longer pulses used here for the purpose of obtaining the stationary response? One potentially useful piece of information to report would be the cooperativity threshold as a function of pulse length. As the authors mention in Line 397, an optically driven maser would indeed be a very useful device; it'd be even more useful if such a maser could operate in a regime of high duty cycle. The data could help the readers access the potentials and limitations of the device in the context of microwave photonics.

We thank the reviewer for pointing out this difference and the future potential for masing. Here we answer in two parts:

- Indeed, in the current manuscript we used longer pulses to verify the coherent stationary electro-optical dynamics. Longer pulses were needed to resolve the spectral shape of the studied effects in sufficient detail. This implied a lower limit on the maximum pump power that can be applied (and thus the C that can be achieved) before the Kerr effect kicks in. However, it is important to note that the cooperativity achieved in this manuscript is already useful for near-term quantum applications, including quantum transduction (the internal conversion efficiency is quite similar for both $C=0.5$ and $C=1$ (89% vs 100%) when compared to other limitations such as the waveguide coupling losses), photon counting protocols (they will in fact **require lower C** to avoid two photon generation), microwave-optics generation ($C < 1$ to stay below the parametric instability regime), and QND measurements of microwaves.

In summary, $C \sim 1$ or $C > 1$ as achieved with shorter pulses is certainly a very interesting new limit for the direct interaction between 2 light waves but in fact not so suitable for most *quantum applications* mentioned above and also results in complicated physics and calibration due to the non-stationary nature when used for such applications.

- The current manuscript focuses on the coherent optical control of superconducting circuits for such quantum applications. Nevertheless, it is one of our set goals to study the masing threshold in detail in the near future – ideally in an optimized device. Optical parametric oscillation (instability threshold) due to the Kerr nonlinearity critically depends on the optical linewidths and the mode configurations [Kippenberg, PRL 2004]. The interplay between χ^2 and χ^3 nonlinearity in the device will enable a wide range of interesting classical applications. For such classical applications like masing via the electro-optical parametric instability, an improvement of the microwave linewidth and the electro-optical coupling will be beneficial for higher cooperativity. Together with thermalization improvements this could lead to significant improvements in terms of duty cycle. In this regard we also note that the current duty cycle is limited by the low trigger rate in order to avoid optical heating for quantum applications. When used as a maser and depending on the application, higher temperatures might be acceptable and a significantly higher duty cycle feasible.

For better clarity, we have rephrased the sentence in lines 292-299.

- Line 375: It may be useful to have a one-sentence explanation on why the Stokes configuration requires a different triggering setting (lower duty cycle).

This difference is not a necessity for the results in the current manuscript but a leftover from other measurements that rely on ultra-low thermal occupancy. While here we focus on optical coherent control of a superconducting cavity, quantum applications, including quantum transduction and microwave-optics entanglement generation, require the operation of the device in the quantum back-action dominant regime, by eliminating the thermal noise, e.g. due to optical heating. This includes the entanglement generation in the Stokes case ($J_s=0$).

To avoid confusion, we have rephrased the sentences in line 383-389.

Finally, some comments and suggestions on semantics / notation / general writing:

- Line 284: The sentence is worded in a somewhat misleading fashion. My understanding is that no data with $C \gg 1$ are provided in the manuscript. The authors may want to refer the reader to their simulation in the SI (Fig. S1) to avoid potential confusion.

We thank the reviewer for the suggestion and have revised the sentence accordingly.

- The notation for various frequencies appear to be used interchangeably from time to time: e.g., Fig. 1c (Ω = optical detuning from each FSR mode), Fig. 2a (Ω = microwave frequency). Although optical detuning is indeed in the GHz regime, my understanding is that the authors intend to reserve Ω (ω) for microwave (optical) frequencies. It would be good to preserve this consistency.

We appreciate the suggestion regarding the consistency of frequencies. We intend to reserve ω for frequencies in the THz range, and Ω for frequencies in the GHz and MHz regime, i.e. both microwave frequencies and frequencies in the rotating frame (susceptibilities in the main text).

- I noticed that some figures are plotted such that data points are cropped out (e.g., Fig. 3b, Fig. 4a). The authors may want to re-adjust the axis limits to include these points fully.

We thank the reviewer for the suggestion and apologize for the partially visible data points, which have been updated in the revised version. More specifically, we have increased the maximum limit of y axis in Fig.3b and x axis in Fig.4a.

We note that, C in the upper panel in Fig. 4a is obtained from the joint fit of the EOIA/EOIT response *within the pump pulse*, i.e. $C > 0$. After the pulse, C decreases to zero, while excess back-action in the microwave cavity still exists, e.g. residual microwave frequency and linewidth changes, as shown in the middle and lower panels in Fig. 4a.

- "Trigger time" is mentioned several times throughout the main text (e.g. Line 375) and the SI. There may be a technical reason for using this term, but I personally find "cycle time" or "repetition time (rate, if quoted in frequency)" to be a little more illuminating.

We thank the reviewer for the suggestion and have changed it to "repetition time" in the revised version.

- Line 400: I am aware that the authors are not claiming to have reached the "quantum-limited regime" in this manuscript and the sentence is meant to be a projection, but I would clearly indicate this as

such. In fact, in their previous paper [3], efforts were made to distinguish between "quantum-limited" and "quantum-enabled".

We thank the reviewer for the suggestion. To avoid confusion, we have rephrased the sentence in line 412.

- Line 394: I imagine many people from the circuit QED community may read this paper. Therefore, it may be helpful to clarify that the "strong interaction regime of microwave photonics" refers to optical generation of microwaves, rather than something involving qubits [e.g. Ma, R., Saxberg, B., Owens, C. et al. *Nature* 566, 51–57 (2019).]

We thank the reviewer for the comment and replace "microwave photonics" with "cavity electro-optics" to avoid confusion among different communities.

- Line 292: I think "Kerr the nonlinearity" should be "Kerr nonlinearity".

We have corrected it.

- I may not be the most qualified to comment on English, but I would use "resonant" ("off-resonance") to replace all the "on-resonant" ("off-resonant").

We thank the reviewer for the comment. Indeed, in the previous version the "on/off-resonant" was used in two different scenarios, i.e. 1) on/off-resonance MW/optical probing, and 2) resonant or off-resonant condition ($FSR =$ or $\neq \Omega_e$). We have revised accordingly to distinguish these two different situations and updated Fig. 4c.

We are also open to suggestions from the copy-editor.

Reviewer #3 (Remarks to the Author):

The authors present results characterizing the electro-optic interaction between a lithium niobate optical resonator and a microwave cavity. They demonstrate phenomena such as EOIA and EOIT, along with dynamical backaction effects, whose analogies were previously mostly observed in optomechanics, but are now accessible in their electro-optic device, thanks to a cooperativity that approaches unity. Furthermore, they present some less well-understood observations, for example a delayed effect of the optical pump on the response of the microwave cavity.

We thank the reviewer for the nice summary of our results, i.e. the coherent optical control of superconducting resonator using dynamical back-action with near unity cooperativity. This is enabled by the minuscule excess back-action in our device, as backed out using a novel and fast coherent measurement technique.

The experimental techniques analysis seem sound, and the results are carefully presented. Given the potential applications of this system, such as quantum transduction, a manuscript presenting the details of its dynamics is certainly useful. However, I am a bit puzzled by the context of this work with respect to previous publications using this device, for example ref. 20 and 22. It seems like the device used here is similar to the one used in those works. The introduction, however, makes it sound like this is a new regime of CEO systems, but the cooperativities shown here were already demonstrated in ref. 20 for example. If there is indeed some new technical development, the authors should make it clearer. If not, the results in this manuscript could have perhaps been included in the supplement to ref. 20 or 22. Though I'm not against publishing a separate paper (but perhaps in a more specialized technical journal) instead of putting all this in a supplement, the relationship to the authors' earlier works should be made clear so as to not confuse the reader.

We appreciate the acknowledgement from the reviewer regarding the scientific rigor and the future applications of our results. Indeed, our result signals the emergence of a mature new platform for **general purpose quantum optical manipulation and readout of microwaves using quantum electro-optics**. It is manifested as first demonstration of electro-optical dynamical back-action, the prerequisite for quantum optical control of microwaves. This differs significantly from earlier experiments, including our earlier work ref. 20 and ref. 22, which only focus on quantum transduction.

- *The feasibility of quantum optical coherent control in the CEO device for wide ranges of applications, either **remains elusive in our previous work, or was even shown impossible repeatedly in other experiments in the community.***
- *Our novel "pump-probe" coherent measurement techniques make it possible to unveil the **coherent temporal and spectral EO dynamics** for the first time.*
- *Our results firmly **cease the long-lasting doubts in the community regarding the compatibility between optical light and microwave superconducting cavity.***
- *Our results are comprehensively presented **under various mode configurations, as required for general-purpose quantum optical coherent control of microwaves.** Ref. 20 (low C, ground state conversion) and 22 (high efficiency and low noise conversion) studied only the microwave-optical transduction using **one subset** of mode configurations, i.e. enhanced anti-Stokes scattering. In both papers, the coherent dynamical back-action leading to the shown effects such as EIT, EIA and light induced spring shifts was not studied. We note that, similar first time demonstrations in optomechanics lead to a number of independent publications.*

- Our results *show the readiness of this approach for a wide range of breakthroughs* that have been anticipated for more than a decade since the original Tsang proposal, e.g. entanglement generation, squeezing, QND measurement, etc., that go ***beyond quantum transduction***.

In our view, once a field (or device platform) is more established and successful, it is also quickly beyond improving only figures of merit (even though that might be an important part of further experiments) and we hope that this will also be the case with cavity quantum electro-optics (as it was the case for circuit QED and cavity optomechanics). Requiring a new device per manuscript implies more interest in the device rather than the physics or the shown application that it enables. In our manuscript we are presenting new physics, and the new material is substantial and can (and in our opinion should) not be covered in the SI of a journal that is already published. Similar observations (e.g. OMIT) at the outset of the field of optomechanics led to a number of separate publications in major journals.

We have revised the abstract, introduction and conclusion to elaborate further the novelty of our results in the revised version.

Some other specific comments are:

1. Under the category of “trapped atoms” on line 57, the authors could consider citing the recent paper by Kumar et al (arXiv:2207.10121).

We have included the suggested reference in the revised version.

2. The authors mention “excess back-action” multiple times in the abstract and intro without explaining what this means. Without going into what exact physical process leads to this back-action, they could at least briefly state what sorts of phenomena they’re referring to (heating, additional loss, etc.)

We thank the reviewer for pointing this out. We do realize that we didn’t introduce excess back-action properly in the previous version. Here it refers to any extraneous perturbations to both optical and microwave modes that are not due to the electro-optical interaction. Such excess back-action can originate from a wide range of phenomena, including piezoelectric, photorefractive, heating, dissipative feedback, quasiparticles effects, etc.

As suggested by the reviewer, we now elaborate further on the possible origins of excess back-action in line 62. In addition, we rephrased the sentences in lines 73-79.

3. I found the labeling of “Stokes” and “anti-Stokes a bit confusing sometimes. In many cases, it would have been more clear if the authors specified where the probe field was located. For example, in the

caption to Fig. 1e, when the authors say the “optical Stokes (orange curve) and anti-Stokes (green curve) responses”, I assume they mean the probe response around mode 1 and 3?

We thank the reviewer for the nice suggestion. For better clarity, we have now added the probing modes labels in the revised version, and updated Fig.1(e) and Fig.3(a).

4. Could the authors comment on the shape of the red curve in the upper right panel of Fig 1e? I guess it's due to some combination of the time-varying narrowing and frequency shift of the mode?

This is correct. The decrease in the *on-resonant* microwave reflection in the beginning of the pulse is mostly due to the narrowing effect while the lateral reflection increase is mostly due to the frequency shift (optical spring effect), which are summarized in the left panel of Fig. 4(a). In addition, as the MW reflection is close to zero (close to critically coupled), the change of R_e (normalized probing field reflection between pulse on and off) can be quite dramatic, as shown in Fig. 1e.

5. In the caption to Fig. 2a, it says “the lower panel shows the reconstructed microwave reflection...” What does “reconstructed” mean?

The “reconstructed” means that in the experiment, the S_{ee} and S_{oo} scattering coefficients are characterized indirectly from the joint fit of normalized reflection $R_{e/o}$ using the time-domain response measurements with swept microwave and optical frequencies. A direct characterization can be rather complicated, due to the complex frequency dependent input and output lines, especially on the optics side due to the optical signal filter (to suppress the strong pump). The detailed data analysis procedure is presented in Section C of SI.

We have reorganized the section for better clarity, as also suggested by Referee 2. In addition, we have rephrased the caption in Fig. 2a.

6. In the discussion of Fig. 2b, the authors say that the deviation of the frequency change is due to small detuning uncertainties. Do they mean the deviation of the data from the theory curve? I guess the theory curve includes the effect of detuning (according to SI A), so does “uncertainty” mean an unknown detuning in the measurement that also seems to vary with cooperativity?

We thank the reviewer for the comment. Indeed, we attribute this to a small experimental uncertainty of the detuning over the long measurement time (9 hours) e.g. for a single blue curve in Fig. 2(b), considering the sweeping of both pump powers and microwave/optical probing frequencies. For simplicity, we assume the same microwave resonance frequency for all measurements over the entire power sweep, even though a slow drift/fluctuation of the device parameters, due to e.g. photorefractive effect on the optical side or quasiparticles on the microwave side may be present.

For better clarity, and as suggested also by reviewer 2,

- 1) we have added additional discussions regarding the discrepancy after line 264.
- 2) we have provided an additional figure in the SI (Section A) to better elaborate on how the microwave frequency changes due to imperfect frequency detunings, i.e. the optical spring effect.

7. In line 375, by "trigger time", do the authors mean repetition period of the experiment?

We have changed this to "repetition time" in the revised version.

REVIEWERS' COMMENTS

Reviewer #2 (Remarks to the Author):

The authors have addressed my comments and updated the manuscript accordingly. The additional experimental details provided are very much appreciated. I recommend publication in Nature Communication.

Reviewer #3 (Remarks to the Author):

I would like to thank the authors for carefully addressing my comments. With the modifications to the manuscript, I think it's clearer that one should see this work demonstrating new phenomena in electro-optics, rather than a more detailed characterization of their transduction device. It is, however, the same device, and my earlier suggestion was to just clearly state this, while saying that now they're using it for a different purpose. I don't think this would detract from the message of the paper, and avoid readers who are familiar with their previous work having to figure out what needed to be different about the physical device parameters in order to perform these experiments.

Regarding the author's point that similar effects in optomechanics were published in major journals, I somewhat agree. On the one hand, demonstrating similar physical phenomena in different physical systems is interesting in general. However, the early results in optomechanics were impactful because before them, we had so few tools for precisely controlling and measuring mechanical objects. The same is not true for microwave resonators. Therefore, I'm not very convinced that this regime is so useful other than for transduction. So, I would agree with referee 1 that this work is borderline in terms of novelty and impactfulness.

I do take some issue with one modification that was made, though: In the conclusions, the authors added a sentence saying that their work "confirms the compatibility between optical light and superconducting microwave circuits". I would argue that that it only confirms the compatibility of optical light with *their particular* superconducting microwave circuit. I'm pretty sure that if I took an arbitrary superconducting circuit and shined some light on it, bad things could still happen.

Reply to Reviewer Comments on Manuscript NCOMMS-22-51829-A

The original referee reports are shown in black. Our replies are shown in blue.

Reviewer #2 (Remarks to the Author):

The authors have addressed my comments and updated the manuscript accordingly. The additional experimental details provided are very much appreciated. I recommend publication in Nature Communication.

We thank the reviewer for the comment and the support of our manuscript for publication.

Reviewer #3 (Remarks to the Author):

I would like to thank the authors for carefully addressing my comments. With the modifications to the manuscript, I think it's clearer that one should see this work demonstrating new phenomena in electro-optics, rather than a more detailed characterization of their transduction device. It is, however, the same device, and my earlier suggestion was to just clearly state this, while saying that now they're using it for a different purpose. I don't think this would detract from the message of the paper, and avoid readers who are familiar with their previous work having to figure out what needed to be different about the physical device parameters in order to perform these experiments.

We thank the reviewer for the comment and the acknowledgement of the new physics we presented. In the revised version, we have included a device characterization section in Methods, where we explicitly state: "The multimode CEO device used in the reported experiments is the one also used in Refs. [20,22]. It consists of"

Regarding the author's point that similar effects in optomechanics were published in major journals, I somewhat agree. On the one hand, demonstrating similar physical phenomena in different physical systems is interesting in general. However, the early results in optomechanics were impactful because before them, we had so few tools for precisely controlling and measuring mechanical objects. The same is not true for microwave resonators. Therefore, I'm not very convinced that this regime is so useful other than for transduction. So, I would agree with referee 1 that this work is borderline in terms of novelty and impactfulness.

We thank the reviewer for this comment and the appreciation of possible new interesting physics with our demonstration. Manifested as the electro-optical dynamical back-action, ***the prerequisite for quantum optical control of microwaves***, our results represent a defining step in the field of cavity electro-optics, which is still in its infancy. Similar to early experiments in the field of optomechanics, our result entails the readiness for exploration of a rich set of exciting physical phenomena that have been anticipated for more than one decade. While most optomechanical systems focus on localized mechanical modes, quantum electro-optics offers physics and applications beyond "microwave resonators", i.e. quantum optical control and measurement of ***microwave fields***, simply due to the propagating nature of the microwave photons. In contrast, propagating mechanical waves (but also resonant mechanical modes) can typically not be measured directly but only via optomechanical scattering.

Below we elaborate further in two parts,

1) the new physics,

Our demonstration of coherent optical control of microwave resonators paves the way for a wide range of experiments, from laser cooling, squeezing, and quantum non-demolition measurement of the microwaves below the standard quantum limit, to microwave-optical entanglement.

In addition to these prospects, abundant unique and unanticipated physics have been already unveiled in our experiments due to our novel measurement techniques, although not explicitly highlighted in the manuscript, e.g.,

- Interpretation of dynamical back-action,

Electro-optical interaction arises between microwave and optical field of distant frequencies. In our analysis, we reveal their interchangeable roles in the interaction, i.e. they are fundamentally equivalent, only constrained by the dissipations.

- Instantaneous dynamical back-action,

Our experiments show for the first time the simultaneous instantaneous temporal and spectral dynamics. To the very best of our knowledge, similar dynamics has never been reported in optomechanics, as most of the experiments focus on the stationary dynamics. Such instantaneous dynamics are critical for future integration with superconducting circuits.

- Normal and reversed dissipation regime,

The dissipations of the microwave (κ_e) and optical (κ_o) modes can be engineered to achieve normal ($\kappa_e < \kappa_o$) and reversed ($\kappa_e > \kappa_o$) dissipation regime. The interchangeable roles in the electro-optical interaction, allows for interesting physics and applications, such as amplification and lasing of both microwave and optical signals, without the need of additional reservoir engineering mechanism, e.g. in Nature Physics 13, 787 (2017).

2) regarding the “impactfulness” of our work,

As mentioned in our previous reply to review 2, the regime is already useful for near-term quantum applications, including quantum transductions, which is currently an important task and a major research topic given the scaling challenges in superconducting quantum computing.

In addition, our result offers a viable route towards a new generation of techniques with simultaneous optical control and readout capabilities of superconducting circuits, benefiting from various mode configurations. For example, this allows for the microwave-optics entanglement generation, Science 380,718 (2023), for future quantum networks of superconducting circuits.

We are convinced that our manuscript will receive immediate interest in the field of cavity electro-optics, nanophotonics, quantum optomechanics, circuit QED, quantum network and etc, and spur new research activities in material sciences and nanotechnology, with platforms facing similar challenges.

I do take some issue with one modification that was made, though: In the conclusions, the authors added a sentence saying that their work “confirms the compatibility between optical light and superconducting microwave circuits”. I would argue that that it only confirms the compatibility of optical light with *their particular* superconducting microwave circuit. I’m pretty sure that if I took an arbitrary superconducting circuit and shined some light on it, bad things could still happen.

We thank the reviewer for the comment. Indeed, over the last years, different research groups in the field have repeatedly shown that light and microwave circuits are incompatible, e.g. due to quasiparticles or excess losses. To avoid confusion, we’ve rephrased this sentence in the conclusion.